# Star-Corrector: A Multi-Turn Interactive Reinforcement Learning Framework for Lean4 Theorem Correction

## Abstract

Formal mathematical reasoning requires models to generate verifiably correct proofs, a process where existing single-turn generation paradigms may fail to utilize the critical feedback from proof checkers. To bridge this gap, we present Star-Corrector (State-Thinking-Answer-Reward Corrector), a multi-turn, feedback-driven framework that explicitly learns from verification signals to refine faulty proofs. Our approach models proof generation as an iterative refinement process: starting from an initial flawed attempt, the model interacts with the Lean verifier at each turn to identify errors and progressively revises the proof. This multi-turn interaction allows the model to internalize corrective signals and learn precise repair strategies. The core contributions of this work are threefold. First, we introduce a multi-turn interaction model that formally defines the proof generation as a process of iterative correction based on verifier feedback, moving beyond single-turn generation. Second, we effectively optimize the policy within this interactive setting by applying GRPO to leverage the sequential verification outcomes. Third, we develop a sampling strategy that dynamically balances problem difficulty during training by leveraging pre-defined difficulty levels and the model's evolving success rate. On the MiniF2F benchmark, STAR-Corrector elevates the pass rate from 64.34% (base model with 32 samples) to 96.72% under a 32+32 sampling budget, marking an absolute gain of +32.37%. The results demonstrate that our approach, particularly the adaptive sampler, effectively enhances generalization on medium and hard problems, validating the importance of closed-loop, data-efficient training for formal reasoning. All code and data are available at an anonymous repository: https://anonymous.4open.science/r/ICLR-Star-E096/.

## 1 Introduction

Automated Theorem Proving (ATP) with LLMs is largely dominated by two technical paradigms. The tree-search approach, pioneered by Jiang et al. (2023), incrementally explores the proof space through step-wise generation and verification. This allows it to discover complex proofs but often at a significant computational cost, with methods like BFS-Prover requiring millions of search operations Xin et al. (2025), and it can lack high-level strategic direction. Conversely, the whole-proof generation paradigm Xin et al. (2024); Polu et al. (2022) produces a proof in a single attempt, leveraging the model's planning capabilities for computational efficiency. However, its one-shot nature lacks intermediate feedback, hindering error correction. Thus, a central challenge remains: effectively integrating the verifiability of interactive tree-search with the strategic planning of whole-proof generation within a unified framework.

To resolve this tension, we introduce a novel paradigm that frames theorem proving as a multi-turn, feedback-driven process of iterative proof refinement. This approach synergizes the verifiability of tree-search with the planning capacity of whole-proof generation by enabling a single model to learn from its mistakes through multiple interactions with the Lean verifier. To effectively optimize this interactive agent, we implement it within the StarPO framework Wang et al. (2025b), which provides a foundation for trajectory-level policy optimization. Our work thus demonstrates a successful instantiation and specialization of this general RL framework for formal theorem proving, offering a principled path to balance planning with verification. In summary, here are our main contributions:

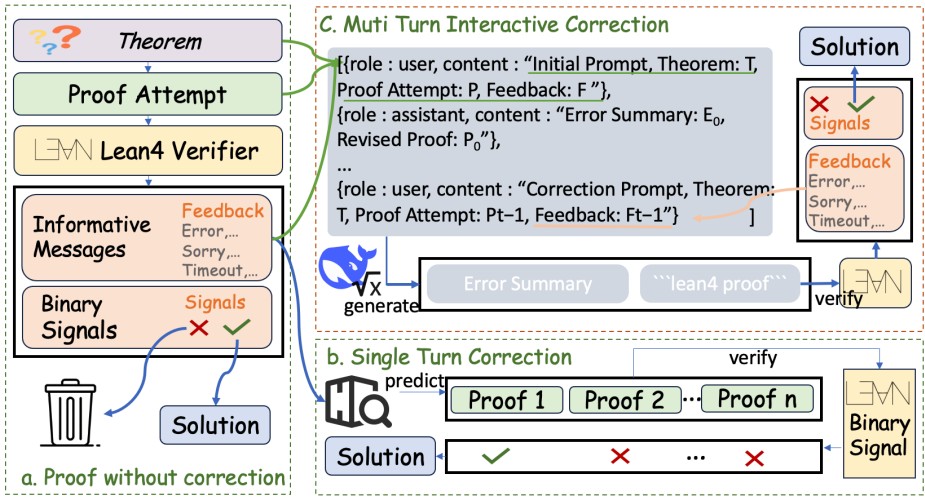

Figure 1: Three approaches to verifier-assisted theorem proof correction: from binary signals to feedback-driven iteration.

1. We propose and implement a multi-turn, feedback-driven proof refinement framework. This framework formalizes the interaction between a prover and the Lean verifier as an iterative process, enabling the model to learn from its mistakes by refining faulty proofs based on verifier feedback.

2. We successfully apply the GRPO algorithm for trajectory-level policy optimization within this interactive setting.

3. We construct a novel, difficulty-stratified dataset for multi-turn theorem proving. And based on it, we develop a dynamic difficulty sampler that adjusts the distribution of training problems in real-time based on the model's overall success rate, ensuring a sustainable learning curve from easy to hard theorems.

4. Our resulting model, Star-Prover, achieves a significant performance improvement, demonstrating a +2% absolute gain in pass rate on the MiniF2F benchmark under a 4-attempt budget, which underscores the effectiveness of our overall training strategy.

## 2    RELATED WORK

**LLM-Based Automated Theorem Proving.** LLM-based ATP has evolved into two main paradigms. One-shot whole-proof generation Xin et al. (2024); Dong & Ma (2025) produces complete proofs in a single attempt, leveraging LLMs' planning capacity but lacking intermediate feedback for error correction. Conversely, step-wise tactical generation Xin et al. (2025) iteratively constructs proofs through verifier-guided search, enabling complex proof discovery but at high computational cost (e.g., millions of expansions Paulson (1994)) and with limited high-level planning. This dichotomy highlights the unmet need to balance efficient planning with interactive verification.

**Reinforcement Learning for Formal Reasoning.** While RL has enhanced NL reasoning by optimizing for final-answer correctness d. Moura & Ullrich (2021), its application to formal ATP faces unique challenges due to sparse rewards. Using Lean's binary pass/fail signal as reward Xin et al. (2024) yields limited gains, underscoring the need for denser training signals. Our work advances this by designing a composite reward function (incorporating temporal discounting and semantic similarity) to provide richer guidance throughout the proof refinement process.

**Multi-Turn Agent Frameworks in LLMs.** General multi-turn agent frameworks Wang et al. (2025b) support trajectory-level optimization for interactive tasks. However, they are not tailored to ATP's deterministic environment and require careful adaptation of state representations and reward structures. Our contribution lies in the effective domain-specific instantiation of such frameworks

for theorem proving, combining GRPO-based optimization with verifier-informed rewards to train a proof refinement agent.

**Curriculum Learning and Adaptive Sampling.** Curriculum learning strategies Jiang et al. (2023) aim to improve training by ordering examples from easy to hard. Traditional methods rely on static difficulty orderings Lightman et al. (2023) or priority buffers Lin et al. (2025), but cannot adapt to model's dynamic capability changes. Our dynamic difficulty sampler addresses this by integrating static difficulty priors with real-time performance signals, ensuring the training distribution continuously matches model capability.

## 3 METHODS

### 3.1 STAR-CORRECTOR FRAMEWORK

**Multi-turn Proof Refinement Framework.** We formulate formal theorem proving as a multi-turn, interactive Markov Decision Process (MDP) between a prover agent and the Lean 4 verifier. As illustrated in Figure 1, our framework operates through an iterative refinement loop: starting from an initial failed proof attempt , the model engages in multiple rounds of interaction with the verifier, progressively correcting mistakes until a valid proof is generated or termination conditions are met.

The core innovation of our approach lies in treating proof generation not as a one-shot task but as a sequential decision-making process. This allows the model to learn from verifier feedback and develop explicit error-correction capabilities. The framework consists of three key components: (1) a state representation that maintains the complete interaction history, (2) an action space that generates structured responses including reflection and proof attempts, and (3) a reward function that provides dense training signals. These components are optimized using Gradient-based Policy Optimization with a dynamic difficulty sampling strategy.

**Problem Formulation as Markov Decision Process.** Following the multi-turn refinement process of Wang et al. (2025b), we model it as a MDP defined by $\mathcal{M} = (\mathcal{S}, \mathcal{A}, \mathcal{R})$, where $S$ represents states , $A$ represents actions, and $R$ denotes the transition dynamics and reward generation process. The agent policy $\pi_\theta$ generates an action $a_t$ at each time step $t$, conditioned on the current state $s_t$ and the interaction history. The environment returns a reward $r_t$ and a new state $s_{t+1}$ given the current transition dynamics:

$$a_t \sim \pi_\theta(\cdot|s_t, \tau_{<t}), \quad (r_t, s_{t+1}) \sim P(\cdot|s_t, a_t), \tag{1}$$

where $\tau_{<t} = \{s_0, a_0, r_0, \ldots, s_{t-1}, a_{t-1}, r_{t-1}\}$ denotes the interaction history. This interactive process continues for a max step $K$, forms a full trajectory $\tau = \{s_0, a_0, r_0, \ldots, s_K\}$ as the materials for continues learning.

**State Space ($\mathcal{S}$).** The state $s_t \in \mathcal{S}$ at turn $t$ encapsulates the *full interaction history* between the prover agent and the Lean 4 verifier, represented as a sequence of structured chat-style messages. This design ensures the agent can reference prior proof attempts, verifier feedback, and self-reflections when formulating revised proofs—critical for tracing errors that persist across multiple turns. Formally, $s_t$ is defined as:

$$s_t = \begin{bmatrix} \{\text{role} : \text{user}, \text{content} : \text{"Initial Prompt, Theorem: } T, \text{Proof Attempt: } P, \text{Feedback: } F\text{"}\}, \\ \{\text{role} : \text{assistant}, \text{content} : \text{"Reflection: } R_0, \text{Error Summary: } E_0, \text{Revised Proof: } P_0\text{"}\}, \\ \vdots \\ \{\text{role} : \text{user}, \text{content} : \text{"Continue Prompt, Theorem: } T, \text{Proof Attempt: } P_{t-1}, \text{Feedback: } F_{t-1}\text{"}\} \end{bmatrix}$$

where:

- $T$ denotes the target theorem (fixed across all refinement turns),
- $P_i$ represents the $i$-th proof attempt (a Lean 4 script),
- $F_i$ denotes the Lean 4 verifier's feedback for $P_i$ (either a success signal or structured error details),
- $R_i$ and $E_i$ correspond to the agent's reflection and error summary for $P_i$, respectively.

The chat-style format is aligned with the input conventions of large language models (LLMs), which excel at modeling contextual sequence data—essential for retaining error-related information across turns. Example prompts are following:

---

**Initial Prompt**

You are an experienced Lean4 expert. Your task is to analyze the given proposition, an incorrect attempt, and its feedback, and then generate a correct solution. Please ensure your response includes the following two parts:
1. **Thought Process**: Detail the error analysis, explain the meaning of the feedback, and reason step-by-step on how to fix the code. This section should showcase your logical reasoning and decision-making process.
2. **Final Answer**: Wrap the final answer in '''lean4. The content inside must include:
- **Problem Analysis**: A brief summary of the core issue or challenge.
- **Core Lemmas**: List the key lemmas or theorems that must be used in the proof (if any), or state that no external lemmas are needed.
- **Core Strategies**: Describe the main sequence of strategies used in the proof and why they are effective.
- **Final Code**: Provide the complete, verifiable Lean4 code.
Base your modifications on the incorrect attempt and the feedback. Ensure the code is correct and efficient.
Proposition: [] Incorrect Attempt Code: [] Feedback: []

---

**Continue Prompt**

The solution you provided in the last response has been tested but unfortunately failed. Latest Test Results: Test Status: Failed Error Message: ' ' Task Instructions: Act as a code debugging expert and carefully analyze the error message above. Please follow these steps:
1. **Review History**: Based on our previous conversation (shown above), analyze how this error is different from or similar to previous ones.
2. **Diagnose the Root Cause**: Precisely explain the meaning of this error message and diagnose the fundamental reason causing it.
3. **Formulate a Solution**: Propose a clear modification plan, stating what needs to be changed and why. Avoid repeating failed approaches from previous attempts.
4. **Generate the Answer**: Finally, provide the complete corrected solution within answer tags. The answer format must include:
- **Problem Analysis**: A brief summary of the current problem.
- **Core Lemmas**: The key lemmas used.
- **Core Strategies**: The sequence of key strategies used.
- **Final Code**: The complete, corrected Lean4 code.
Please begin your analysis.

---

**Action Space** ($\mathcal{A}$). An action $a_t \in \mathcal{A}$ is the agent's structured response to the current state $s_t$, designed to mandate explicit reasoning prior to proof revision. Each $a_t$ comprises two mandatory, mutually complementary components:

1. **Chain-of-Thought (CoT) Reflection**: A natural-language analytical account of why the previous proof attempt $P_t$ failed. This component links specific errors in $P_t$ to underlying logical or syntactic flaws.

2. **Revised Proof Code**: A complete Lean 4 script $P_{t+1}$ that incorporates corrections for the issues identified in the error summary. Incomplete or syntactically invalid code (e.g., truncated tactic sequences, unclosed brackets) is classified as a format error, ensuring only verifiable scripts are submitted to the Lean 4 checker. Detailed settings are in Appendix A

This structured action space enforces that every revision is grounded in explicit reasoning, reducing the probability of introducing new errors during refinement.

**Reward Function.** Unlike stepwise reward schemes that provide feedback at each intermediate step, our reward function assigns feedback at the trajectory level, that is an entire sequence of proof at-

tempts is compiled and verified by the Lean 4 system. Thus, the *total reward* synthesizes these three complementary components to balance competing objectives: (1) *format reward* the presence of complete Lean 4 code, (2) *validity reward*incorporating a temporal discount factor based on the number of attempts required for successful verification by Lean 4, and (3) *weighted similarity reward* semantic similarity to a reference solution. The precise formulation of *total reward* as a weighted combination of these components, including tunable hyperparameters for balancing priorities, is detailed in Appendix A.

**Policy Optimaization Algorithms.** We employ the StarPOWang et al. (2025b) as our RL algorithm, we assign a scalar reward $R(\tau_i)$ to each trajectory and normalized advantage $\hat{A}_{i,t}$ across all tokens in $\tau_i$:

$$\hat{A}_{i,t} = \frac{R(\tau_i) - \text{mean}(\{R(\tau_1), \dots, R(\tau_G)\})}{\text{std}(\{R(\tau_1), \dots, R(\tau_G)\})}. \tag{2}$$

The GRPO objective becomes:

$$J_{\text{GRPO}}(\theta) = \frac{1}{G} \sum_{i=1}^{G} \frac{1}{|\tau_i|} \sum_{t=1}^{|\tau_i|} \min \left[ \frac{\pi_\theta(\tau_{i(t)} \mid \tau_{i,<t})}{\pi_{\text{old}}(\tau_{i(t)} \mid \tau_{i,<t})} \cdot \hat{A}_{i,t}, \ \text{clip}\left( \frac{\pi_\theta(\tau_{i(t)} \mid \tau_{i,<t})}{\pi_{\text{old}}(\tau_{i(t)} \mid \tau_{i,<t})}, 1 - \varepsilon, 1 + \varepsilon \right) \cdot \hat{A}_{i,t} \right]. \tag{3}$$

$\varepsilon$ is a hyperparameter. In our experiments, we set $\varepsilon = 0.2$. This clip ratio setting enables stable policy updates.

## 3.2 DATA CURATION

**Data Curation.** We construct a large-scale dataset for training proof correction models through a multi-stage, systematic generation process. Our pipeline begins with a curated set of 9,000 high-quality theorems from the NuminaMath-Lean Wang et al. (2025a) corpus. To establish a reliable difficulty metric, we leverage a powerful prover model to generate eight distinct proof attempts for each problem, categorizing them into difficulty tiers based on the empirical success rate. This process yields over 72,000 proof-generation instances, providing a robust foundation for difficulty assessment. We then employ DeepSeek Reasoner DeepSeek (2025) as a teacher model to simulate authentic correction trajectories. For each problem, the model engages in a multi-turn interaction following the STAR-corrector protocol, starting from an initial erroneous proof and iteratively refining it based on granular feedback from the Lean 4 verifier. This procedure generates a rich dataset of over 2477 interactive correction trajectories. The final training set is distilled from these trajectories by selecting only the successful sequences where the model ultimately produced a verified correct proof, resulting in a high-quality corpus of 918 problem-correction pairs. This dataset specifically captures the dynamic process of debugging and refinement, making it ideally suited for training models to diagnose errors and implement corrective actions in mathematical theorem proving.

## 3.3 DYNAMIC DIFFICULTY-AWARE SAMPLER

We implement a dynamic weight adjustment mechanism to align training data distribution with the model's evolving capabilities: First, from our proposed multi-turn-correction datasets, we re-grade problems into multi-turn-correction Difficulty levels based on their original difficulty grades and the step number of the teacher model (DeepSeek-R1 DeepSeek (2025)) when successfully generating proofs; sampling weights for these 5 levels are then dynamically adjusted based on the smoothed global pass rate over recent steps, with stage-based shifts in emphasis across levels, and weights normalized to form a valid distribution. Detailed difficult and numerical settings are in TableA.6.

## 4 EXPERIMENTS

**Base Model.** For our work, we select DeepSeek-Prover-V2 7B Z.Z. Ren (2025) as the base model, with considerations rooted in its prominent advantages aligned with our research needs. Specifically, this model first demonstrates strong performance on existing benchmarks. Second, it supports extended context length enabling processing of long-chain logical dependencies in multi-step proofs. Third, it is equipped with a high-precision Chain-of-Thought (CoT) mode, which elaborates intermediate reasoning steps to enhance the transparency of formal proof generation . Most importantly,

its core recursive theorem decomposition mechanism not only mimics human-like problem-solving and reduces computational costs, but also aligns with our "State-Thinking-Actions-Reward" Wang et al. (2025b) modeling process.

**Implementation.** We conducted the RL training phase on a computational node equipped with 4×NVIDIA A100 80GB GPUs. At each training step, we dynamically sample a total of 128 "Theorem–Bad Attempt–Feedback" pairs from various difficulty levels. For each such pair, we generate 8 proof trajectories, resulting in a global batch size of 128×8=1024 trajectories. The maximum number of turns per trajectory is set to 4, and the maximum response length is limited to 3000 tokens for each turn. A rollout ratio of 0.3 is applied to retain the highest-value trajectories for further optimization. It is crucial to note that the rollout process—which involves generating these 1024 trajectories, each requiring up to 4 synchronous calls to the Lean 4 verifier—constitutes the primary computational bottleneck of our framework. This significant resource demand, particularly for the verifier interactions, currently limits the ease of reproduction for researchers without access to comparable large-scale computational resources. The resulting model obtained after this RL training phase is named Star-Corrector.

**Benchmarks.** Following previous workZ.Z. Ren (2025); Lin et al. (2025); Yu et al. (2025), we select the most commonly-used MiniF2F-test Zheng et al. (2022) to validate the effectiveness of our proposed method. Due to computational constraints, we adjusted the configuration of DeepSeek Prover V2 by limiting the proof generation length to 8192 tokens and reducing the Lean4 verifier timeout from 300 to 60 seconds, balancing experimental efficiency and practicality. Under this setup, we establish baseline experiments with sampling budgets of 1 (greedy decoding) and 32 (multi-sample decoding) to provide a rigorous benchmark. Our core objective is to derive valid proofs by correcting flawed proof attempts. To this end, we adopt a validation protocol that involves appending corrections to the flawed proofs generated by the baseline model. We quantitatively evaluate pass accuracy under different sampling budgets-before and after applying our correction.

Table 1: Comparison with state-of-the-art methods on the miniF2F-test dataset.

| Method | Sample budget | miniF2F-test |
|---|---|---|
| TheoremLlamaWang et al. (2024) | 128 | 33.6% |
| DeepSeek-Prover-v1.5-RLXin et al. (2024) | 32 | $50.0 \pm 0.5\%$ |
| | 64 | $50.7 \pm 0.4\%$ |
| | 128 | $51.6 \pm 0.5\%$ |
| | 3200 | $54.9 \pm 0.7\%$ |
| Goedel-Prover-SFTLin et al. (2025) | 32 | $57.6 \pm 0.7\%$ |
| | 3200 | 62.7% |
| DeepSeek-Prover-V2-7B(CoT)Z.Z. Ren (2025) | 1 | 58.6% |
| | 32 | 75.6% |
| DeepSeek-Prover-V2 (Unified Experimental Setup) | 1 | 53.12% |
| | 2 | 58.4% |
| | 32 | 64.34% |
| DeepSeek-Prover-V2 + StarCorrector | 1+1 | **58.04%** |
| | 32+32 | **96.72%** |

**Results on MiniF2f.** The primary evaluation metric was pass rate—the percentage of theorems with verified correct proofs—under varying sampling budgets. Our framework's performance is reported as "base model + Star-Corrector" (e.g., 1+1 denotes 1 sample for the base model's initial attempt and 1 sample for Star-Corrector's correction). Based on the results presented in TABLE1, we observe a nuanced relationship between sampling budget and the effectiveness of the STAR-Corrector framework. Specifically, under low sampling budgets, the 1+1 correction (58.04%) demonstrates a clear improvement over the base model with 1 sample (53.12%), achieving an absolute gain of 4.92%. However, this corrected performance does not surpass the base model's result with 2 samples (58.4%). This indicates that while a single refinement turn can rectify critical errors in the initial attempt, its corrective power is constrained when compared to simply generating multiple independent proofs without iterative feedback. The limited diversity in greedy decoding (1 sam-

ple) restricts the scope for error correction, as the model has fewer alternative paths to explore and refine. In contrast, under high sampling budgets, the synergy between multi-sample decoding and multi-turn correction becomes markedly more effective. The 32+32 configuration achieves a pass rate of 96.72%, significantly outperforming the base model with 32 samples (64.34%) by an absolute margin of 32.37%. This substantial improvement underscores that as the base model produces a wider variety of initial proof attempts, STAR-Corrector's iterative refinement mechanism—guided by MDP-based state tracking and GRPO-optimized policy—becomes increasingly impactful. The framework efficiently filters out invalid paths, leverages verifier feedback across turns, and converges to valid proofs more rapidly, highlighting the compounded benefits of planning and verification in a multi-turn setting.

## 5 CONCLUSION

In this work, we introduced STAR-Corrector, a multi-turn, feedback-driven framework for automated theorem proving that bridges the gap between the strategic planning of whole-proof generation and the verifiability of interactive tree-search. By formalizing proof refinement as an interactive Markov Decision Process and optimizing the agent with GRPO under a composite reward function, our approach enables the model to learn from its mistakes through iterative interactions with the Lean 4 verifier. Furthermore, we introduced a dynamic difficulty-aware sampling strategy that adapts the training distribution in real-time to match the model's evolving capabilities. Experimental results on the MiniF2F benchmark demonstrate that STAR-Corrector consistently improves upon a strong baseline—DeepSeek-Prover-V2—across sampling budgets. Notably, under a high sampling budget (32+32), our method achieves an absolute gain of +6.97% in pass rate, highlighting the synergistic effect of combining multi-sample decoding with multi-turn refinement. Even under a low budget (1+1), the framework yields a meaningful improvement, illustrating its ability to leverage verifier feedback for error correction even when exploration is limited.

## 6 LIMITATIONS

Our approach has several limitations. First, the performance of STAR-Corrector is inherently dependent on the quality and diversity of the initial proof attempts generated by the base model. If the base model fails to produce a minimally plausible starting point, the correction process may struggle to recover. Second, the training process is computationally intensive, primarily due to the massive rollout requirements for generating interaction trajectories. Each trajectory involves multiple turns of proof generation and, crucially, synchronous verification by the Lean 4 verifier. This extensive interaction with the verifier during the data collection phase constitutes the primary computational bottleneck, potentially limiting the accessibility and reproducibility of our method for researchers with limited resources. Third, the current framework is constrained by a fixed maximum number of refinement turns (set to 4 in our experiments), which may be insufficient for highly complex theorems requiring deeper iterative debugging. Finally, our reward function, while carefully designed, still relies on handcrafted components and hyperparameters, which may not generalize optimally across all theorem distributions or proof styles.

## 7 ETHICS STATEMENT

For ethics compliance, this work uses 9,000 theorems from the open-licensed NuminaMath-Lean corpus Wang et al. (2025a) and 2000+ synthetic proof correction trajectories generated via DeepSeek Reasoner's public API (in line with its terms of service); all data is anonymized in the anonymous repository (https://anonymous.4open.science/r/ICLR-Star-E096/) to meet ICLR 2026's double-blind review policy, with full licensing details stored there. STAR-Corrector accelerates mathematical research and supports Lean4 education through interpretable error analysis, while risks like misinformation are mitigated by requiring Lean4 verification for all valid proofs, framing the model as a human-collaborative tool, and optimizing training (e.g., 4-turn refinement limit, 60s verifier timeout) to reduce energy use; its design combining Chain-of-Thought reflection and Lean4 code ensures traceable revisions, with all results relying on deterministic Lean4 verification and full training logs in the repository for independent validation.

## 8 REPRODUCIBILITY STATEMENT

To ensure the reproducibility of the work presented, we have made extensive efforts documented in the main paper, the appendix, and the supplementary materials. The Lean4-formatted MiniF2F-test benchmark files are available in an anonymous repository at https://anonymous.4open.science/r/ICLR-Star-E096/, with post-review plans to migrate to a permanent, DOI-linked platform. Key experimental configurations necessary for replication—such as the DeepSeek-Prover-V2 7B base model selection, global batch size of 1024, maximum 4 refinement turns per problem, and 60s Lean4 verifier timeout—are detailed in Section 4 (Experiments), while the implementation of the STAR-Corrector framework (state representation, action space, reward function) and dynamic difficulty sampler (difficulty grading rules, weight adjustment logic) are described in Sections 3.1 and 3.3, respectively. Appendix A provides complete definitions of the composite reward function (format, validity, and weighted similarity components) and default hyperparameters, and supplementary materials in the repository include a troubleshooting guide for common experimental issues.

## A DETAILED DEFINITION OF THE REWARD FUNCTION

### A.1 FORMAT REWARD

The format reward enforces the prerequisite of verifiable code by penalizing non-complete outputs. It is defined as a binary scalar to prioritize syntactic well-formedness:

$$R_{\text{format}} = \begin{cases} 0.5 & \text{if code\_exist} = 1 \quad \text{(complete Lean 4 code present)}, \\ -1 & \text{if code\_exist} = 0 \quad \text{(no complete Lean 4 code)}. \end{cases}$$

- The positive value (0.5) incentivizes outputs that can be processed by the Lean 4 verifier. - The negative penalty ($-1$) discourages non-verifiable outputs (e.g., partial code, natural language only), which cannot contribute to logical validation.

### A.2 VALIDITY REWARD

The validity reward incentivizes two objectives: *logical correctness* (via Lean 4 verification) and *iteration efficiency* (via minimizing the number of attempts). It is non-zero **only if** the proof passes verification:

$$R_{\text{validity}} = \begin{cases} 1 + \frac{T_i - M}{2(T_i + M)} & \text{if pass} = 1 \quad \text{(proof validated)}, \\ 0 & \text{if pass} = 0 \quad \text{(proof invalid)}. \end{cases}$$

Where:

- $T_i$: Total number of attempts allocated to the target theorem by the Deepseek reasoner DeepSeek (2025);
- $M$: Actual number of attempts used to reach the valid proof (only relevant when pass $= 1$).

The term $\frac{T_i - M}{2(T_i + M)}$ introduces an **efficiency bonus** (ranging from 0.7 to 1.3). Shorter trajectories (smaller $M$) yield higher rewards, encouraging the policy to avoid redundant iterations and converge to valid proofs faster.

### A.3 WEIGHTED SIMILARITY REWARD

The weighted similarity reward guides the policy toward human-interpretable reasoning by scaling $R_{\text{sim}}$ with the verification status. The similarity reward $R_{\text{sim}}$ is quantified by the **normalized edit distance** between the generated proof and the reference proof. This balances interpretability for valid proofs and exploratory flexibility for invalid attempts:

$$R_{\text{sim, weighted}} = \begin{cases} \alpha \cdot R_{\text{sim}} & \text{if pass} = 1 \quad \text{(emphasize similarity for valid proofs)}, \\ \gamma \cdot R_{\text{sim}} & \text{if pass} = 0 \quad \text{(weaker guidance for invalid attempts)}. \end{cases}$$

Table 2: Default Hyperparameters for the Reward Function

| HYPERPARAMETER | VALUE | PURPOSE |
| --- | --- | --- |
| $\alpha$ | 0.3 | Weight for similarity reward in valid proofs |
| $\gamma$ | 0.3 | Weight for similarity reward in invalid proofs |
| $\beta$ | 0.5 | Weight for format reward in invalid code-complete proofs |
| $\tau$ | 0.5 | Reward cap for Subcase 1b (invalid code-complete proofs) |

Where $\alpha, \gamma \in (0, 1)$ are tunable hyperparameter: - Example configuration: $\alpha = 0.8$ (prioritizes alignment with reference solutions in educational benchmarks, where interpret-ability is critical); - Example configuration: $\gamma = 0.3$ (provides light guidance for invalid attempts to avoid over-constraining exploration of alternative proof paths).

### A.4 TOTAL REWARD CALCULATION RULES

The total reward ($R_{\text{total}}$) integrates the three components and applies a cap ($\tau$) to prevent over-inflation of non-valid outcomes. It is structured around the presence of complete Lean 4 code (i.e., code_exist), as this determines the feasibility of further validation.

#### A.4.1 CASE 1: COMPLETE LEAN 4 CODE EXISTS

For verifiable code, the total reward prioritizes logical validity while accounting for format and similarity.

**Subcase 1a: Proof is Validated (pass = 1)** Valid proofs inherently satisfy the format requirement (by code_exist = 1), so the format reward is omitted to prioritize validity and interpret-ability:

$$R_{\text{total}} = R_{\text{validity}} + R_{\text{sim, weighted}}.$$

**Subcase 1b: Proof Fails Verification (pass = 0)** For code-complete but logically invalid attempts, a weight $\beta \in (0, 1)$ balances recognition of format compliance with penalties for logical errors. A reward cap ($\tau$) prevents overvaluing partial progress:

$$R_{\text{total}} = \beta \cdot R_{\text{format}} + R_{\text{sim, weighted}} \quad (\text{capped at } \tau).$$

Example configuration: $\beta = 0.6$ (moderate recognition of format compliance without neglecting logical flaws).

#### A.4.2 CASE 2: NO COMPLETE LEAN 4 CODE

Non-verifiable outputs (e.g., truncated code, natural language) contribute no actionable progress to proof correction. The total reward reduces to the format penalty:

$$R_{\text{total}} = R_{\text{format}} = -1.$$

### A.5 HYPERPARAMETER CONFIGURATION

For reproducibility, the default hyperparameters used in the experiments are listed in Table A.5.

### A.6 DIFFICULTY CLASSIFICATION

- **Easy**: Pass@8 = 1 (the SFT model consistently generates valid proofs in one turn),

- **1**: Pass@8 > 0.75 (the SFT model requires 2–3 refinement turns to generate valid proofs),

- **2**: $0.50 \leq$ Pass@ $\leq 0.75$ (the SFT model requires 2–3 refinement turns to generate valid proofs),

- **3**: $0 \leq$ Pass@8 $\leq 0.5$ (the SFT model requires 2–3 refinement turns to generate valid proofs),
- **4**: Pass@8 $=< 0.$ (the SFT model struggles to generate valid proofs even with multiple refinement turns).

## A.7 THE USE OF LARGE LANGUAGE MODELS (LLMS)

We acknowledge the use of large language models (LLMs) in the preparation of this work. Specifically, we used Doubao ByteDance (2025) and DeepSeek Reasoner DeepSeek (2025) to assist with refining the language expression of the manuscript (e.g., improving clarity and flow of text) and to help draft an initial outline of the related works section by summarizing key findings from a set of pre-identified relevant papers. All content, including the final version of the related works and refined text, was critically reviewed, edited, and approved by the authors to ensure accuracy, originality, and alignment with the research contributions of this work.

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
