# OpenReview forum: "Star-Corrector: A Multi-Turn Interactive Reinforcement Learning Framework for Lean4 Theorem Correction"
_ICLR.cc/2026/Conference — Submitted to ICLR 2026_

### Official Review · Reviewer_39LF · 2025-10-26

**Soundness:** 2
**Presentation:** 2
**Contribution:** 2
**Rating:** 4
**Confidence:** 2

**Summary:**

This work presents a model trained as an iterative corrector for automated theorem proving. The model operates in a multi-turn process, interacting with the Lean verifier to identify and progressively fix errors in a proof. The experimental results are impressive: when combined with Deepseek-prover-V2, the corrector achieves a very high pass rate on the miniF2F benchmark.

**Strengths:**

Regarding originality, although several concurrent works (e.g., Goedel-Prover-v2, Seed-Prover) are also exploring the idea of multi-turn revision for ATP, the concept in this paper is still valuable (since previous works like Deepseek-Prover-v2 and Goedel-Prover did not explore this direction). Regarding significance, the reported performance is truly impressive, demonstrating the effectiveness of their method.

**Weaknesses:**

The first weakness is that the reported results seem questionably high, perhaps "too good to be true." This work claims 96% on miniF2F with a "32 + 32" budget (which is not clearly defined as an equivalent pass rate), a result significantly stronger (over 30%) than Deepseek-prover-v2. Moreover, this performance is reportedly achieved with an extremely small training dataset. Section 3.2 states the high-quality corpus consists of only 918 problem-correction pairs. This seems implausibly small, as previous or concurrent works like Deepseek-prover and Goedel-prover use datasets on the order of 100K to 1M for the SFT phase and over 10K for the RL phase.

The second issue is that many experimental descriptions are very unclear. It is not explained what a "32 + 32" budget means or what its equivalent pass@k configuration is. Also there are contradictory gains reported: line 328 claims a 32.37% gain, but line 345 claims a 6.97% gain. It is not clear where these numbers come from or what they are being compared against.

The third point is that the experimental setup at line 309 appears to contradict the results from the original Deepseek-Prover-V2 paper. This work states the baseline pass@32 performance is only 64% under their "unified experimental setup," whereas the original Deepseek-Prover-V2 paper reported a performance of over 80%.

**Questions:**

1. Could you please clarify the "32+32" experimental setup? What does this configuration specifically entail (e.g., number of samples, correction rounds), and what is its equivalent pass@k metric?

2. We noted a discrepancy in the Deepseek-prover-v2 baseline results: the performance cited in line 310 is significantly lower than that reported in the original paper. What specific factors in your "unified experimental setup" cause this mismatch? Were you able to successfully reproduce the original paper's reported results using your infrastructure?

3. Could you please specify which base model (e.g., architecture and pre-trained weights) was used to train your corrector?

---

### Official Review · Reviewer_zBW5 · 2025-10-31

**Soundness:** 2
**Presentation:** 1
**Contribution:** 2
**Rating:** 2
**Confidence:** 4

**Summary:**

This paper aims to define the task of ATP as "an iterative refinement process: starting from an initial flawed attempt, the model interacts with the Lean verifier at each turn to identify errors and progressively revises the proof." The paper experiments on miniF2F dataset using DeepSeek Prover V2.

**Strengths:**

I can't think of any strengths for the current version of the paper.

**Weaknesses:**

Token budgets are not reported for the experiments. With no information about the token budget, it is difficult to evaluate the contribution of the paper.


The method is not described clearly enough. Very little information is provided about the method and the experiments while page 4 is mostly dedicated to prompt templates and the paper concludes on the 7th page.

It is not clear to me what 32 + 32 means when the paper reports it for it s sampling budget. How should one compare the 32 budget to a 32 + 32 budget?

Paper talks about curating a training set but it remains unclear to me how the training set is used for training purposes.

Results in Table 1 are very brief and the jump in accuracy comes with almost no explanation or additional details about how it has been achieved.

Given the paper's view of proof trajectories, I would have guessed that it will use a step prover model and also utilize the tree search methods. However, the paper uses whole proof generation models such as DSP V2. There is also no discussion of step provers such BFS Prover, InternLM, etc. I'm not sure if this is because of lack of familiarity with the existing models or not.


Citations mostly use wrong formatting. This is more troublesome in the tables.

**Questions:**

Any clarifications may be helpful.

---

### Official Review · Reviewer_Q195 · 2025-11-01

**Soundness:** 3
**Presentation:** 2
**Contribution:** 2
**Rating:** 4
**Confidence:** 4

**Summary:**

Authors propose reframing automatic theorem proving as a iterative refinement process and using trajectory-level RL to optimize on these longer horizons.

Contributions include:
- multi-turn formulation that incorporates lean verifier feedback
- trajectory level GRPO RL
- curriculum sampling: dynamically adjusts sampling weights for problem difficulty strata

The authors show improved pass rates on miniF2F test split.

**Strengths:**

- formalization of proof repair as MDP
- strong results (improvement) on miniF2F, a standard benchmark for automatic theorem proving work
- curriculum learning approach in RL loop

**Weaknesses:**

- iterative self repair for proof generation is not entirely novel. [Zhou et al 2025](https://arxiv.org/abs/2507.15225)'s "Solving Formal Math Problems by Decomposition and Iterative Reflection" have propose something similar and additionally add complexity with problem decomposition, achieving similar miniF2F score (95) without training (not cited)
  - I think Goedel Prover also uses proof repair as a training objective?
- lack of a training-compute equalized baseline-- would be instructive to compare this training approach to single turn proof generation
- little analysis of the impact of the curriculum sampling

**Questions:**

- What exactly is meant by the "x+y" sampling budget?
- Were there any ablation studies on the curriculum sampling component?
- Are other benchmarks evaluated?

---

### Official Review · Reviewer_kmCC · 2025-11-03

**Soundness:** 1
**Presentation:** 1
**Contribution:** 1
**Rating:** 0
**Confidence:** 5

**Summary:**

The paper introduces STAR-Corrector, a multi-turn reinforcement learning framework for Lean4 theorem correction. Unlike single-turn proof generation, STAR-Corrector models theorem proving as an iterative dialogue between the model and the Lean verifier. Each turn involves (1) analyzing verifier feedback, (2) generating a reflection on errors, and (3) proposing a corrected proof attempt. The agent is trained using GRPO with a composite reward that combines verification success, efficiency (fewer attempts), code validity, and similarity to reference proofs. A dynamic difficulty-aware sampler adjusts the problem distribution based on model progress.

**Strengths:**

- The proposed method is intuitive and reasonable.

**Weaknesses:**

The paper is poorly written, with several significant issues, including:

- Numerous formatting errors. For example, references in Table 1 are not cited in the correct format, and the authors misuse \citep and \citet throughout the paper. Additionally, the appendix appears immediately after the main text instead of following the references.

- Inconsistencies in reported results. In Table 1, the authors report an absolute gain in accuracy of 32.37%, whereas line 345 states, “our method achieves an absolute gain of +6.97% in pass rate.”

- Missing technical details. For instance:
    - In line 259, the authors write, “sampling weights for these 5 levels are then dynamically adjusted based on the smoothed global pass rate over recent steps,” but the procedure is not clearly explained.
    - In line 478, the authors state, “For reproducibility, the default hyperparameters used in the experiments are listed in Table A.5.” However, Table A.5 does not exist.
    - A link to an anonymous repository is provided, but the repository is empty.
    - DeepSeek-Prover-V2 (Unified Experimental Setup) is listed as a main baseline, but the paper does not explain what this setup is.
    - Given that the main content ends halfway through page 7, the authors should have sufficient space to include these missing details.

Due to the numerous writing issues and missing information, I am unable to assess the quality of the reported results. I believe this paper requires substantial revision.

**Questions:**

See weaknesses.

**Details Of Ethics Concerns:**

Given the way this paper is written, I am skeptical of the reported results' authenticity.

---

### Meta-Review · Area_Chair_RvBW · 2026-01-06

**Summary:**

Concerns raised by reviewers:
- severe formatting errors and contradictory results (+32.37% vs +6.97% gains) per kmCC;
- unclear experimental setup, especially the undefined "32+32" sampling budget (zBW5, 39LF, Q195);
- missing hyperparameters, token budgets, and method details (kmCC, zBW5);
- implausibly high performance (96.72% on MiniF2F) from only 918 training pairs (39LF);
- mismatched DeepSeek-Prover-V2 baselines (39LF); questionable novelty given concurrent work (Q195);
- and an empty anonymous repository (kmCC).

**Reviewer Concerns:**

Not applicable since no author responses are made.

**Reviewer Scores:**

Not applicable since no author responses are made. The original scores are 0, 4, 2, 4.

---

### Decision · Program_Chairs · 2026-01-26

Reject